# Host, parasite and drug determinants of clinical outcomes following treatment of visceral leishmaniasis: a protocol for individual participant data meta-analysis

Rishikesh Kumar [1], Prabin Dahal [2,3], Sauman Singh-Phulgenda [2,3], Niyamat Ali Siddiqui,[1] Abdalla Munir [2,3], Caitlin Naylor,[2,3] James Wilson [2,3], Gemma Buck,[2,3] Manju Rahi,[4] Paritosh Malaviya,[5] Fabiana Alves,[6] Shyam Sundar,[5] Koert Ritmeijer,[7] Kasia Stepniewska,[2,3] Philippe J Guérin,[2,3] Krishna Pandey [1]

**Correspondence to**
Prabin Dahal;
prabin.dahal@ndm.ox.ac.uk and Dr Krishna Pandey;
drkrishnapandey@yahoo.com

## ABSTRACT

**Introduction** Visceral leishmaniasis (VL) is a parasitic disease with an estimated 30 000 new cases occurring annually. There is an observed variation in the efficacy of the current first-line therapies across different regions. Such heterogeneity could be a function of host, parasite and drug factors. An individual participant data meta-analysis (IPD-MA) is planned to explore the determinants of treatment outcomes.

**Methods and analysis** The Infectious Diseases Data Observatory (IDDO) VL living systematic review (IDDO VL LSR) library is an open-access resource of all published therapeutic studies in VL since 1980. For this current review, the search includes all clinical trials published between 1 January 1980 and 2 May 2021. Studies indexed in the IDDO VL LSR library were screened for eligibility for inclusion in this IPD-MA. Corresponding authors and principal investigators of the studies meeting the eligibility criteria for inclusion were invited to be part of the collaborative IPD-MA. Authors agreeing to participate in this collaborative research were requested to share the IPD using the IDDO VL data platform. The IDDO VL data platform currently holds data sets from clinical trials standardised to a common data format and provides a unique opportunity to identify host, parasite and drug determinants of treatment outcomes. Multivariable regression models will be constructed to identify determinants of therapeutic outcomes using generalised linear mixed-effects models accounting for within-study site clustering.

**Ethics and dissemination** This IPD-MA meets the criteria for waiver of ethical review as defined by the Oxford Tropical Research Ethics Committee (OxTREC) granted to IDDO, as the research consists of secondary analysis of existing anonymised data (Exempt granted on 29 March 2023, OxTREC REF: IDDO) Ethics approval was granted by the ICMR-Rajendra Memorial Research Institute of Medical Sciences ethics committee (Letter no: RMRI/EC/30/2022) on 04-07-2022. The results of this IPD-MA will be disseminated at conferences, IDDO website and any peer-reviewed publications. All publications will be open source.

## STRENGTHS AND LIMITATIONS OF THIS STUDY

⇒ In any single visceral leishmaniasis (VL) clinical trial, only few relapses are observed which limits the ability to identify predictors associated with it. An individual participant data meta-analysis (IPD-MA) will increase the statistical power to detect the predictors and moderators of treatment outcome.

⇒ The identification of studies eligible for inclusion in the IPD-MA has been made through a comprehensive literature search of all published studies since 1980 with predefined inclusion–exclusion criteria. However, retrieval of data from trials published prior to 2000 can be a major challenge.

⇒ This IPD-MA will use the VL repository of IPD hosted by Infectious Diseases Data Observatory (The IDDO VL data platform). A major strength of this study is that data on the IDDO VL data platform is harmonised to a common standard based on an extensive consultation with the VL research community.

⇒ A particular scientific challenge is that distinction of relapse from reinfection is seldom carried out in VL therapeutic trials and hence would not be considered in this analysis.

Findings of this research will be critically important for the control programmes at regional/global levels, policy makers and groups developing new VL treatments.
**PROSPERO registration** CRD42021284622.

## INTRODUCTION

Visceral leishmaniasis (VL), also known as kala-azar, is a parasitic disease with anthroponotic and zoonotic modes of transmission.[1] The disease is characterised by prolonged fever, cachexia, splenomegaly, hepatomegaly and anaemia. The annual global incidence of VL is estimated at 30 000 cases, with only 25%–45% of the cases reported to the WHO.[2]

The disease has an outbreak potential and linked with war, conflicts and climate change, and if left untreated, the disease is fatal in over 95% of the cases.[2–4]

The efficacy of antileishmanial regimens in VL varies across geographical regions. For example, single-dose liposomal amphotericin B (L-AmB) has demonstrated a high efficacy (≥94%) in the Indian subcontinent while its efficacy is suboptimal (58%) in East Africa.[5 6] There are high levels of resistance against antimony-based drugs in India, although it continues to be used in a combination regimen with paromomycin as a first-line therapy in East Africa.[5 6] The underlying reasons for such variation are not fully understood but the observed therapeutic response is likely multifactorial and may be a function of drug resistance, variation in the pharmacokinetic and pharmacodynamic properties of the drugs used and the underlying host immunity.[7–10]

Several studies have reported on the determinants of therapeutic outcomes in studies conducted across different geographical regions and patient populations.[11–16] Additionally, the heterogeneity introduced by variation in study design and conduct leads to challenges in reliably comparing the effect measures across the studies. A major challenge in reliably assessing risk factors of therapeutic outcomes is that at individual study level, the number of relapses is often small,[17] which limits the precision of the estimated effect. Individual participant data (IPD) meta-analysis (IPD-MA) of existing studies can overcome this limitation by increasing the effective sample size and allows for a greater power in detecting differential treatment effects across individuals in trials.[18]

### Objectives
The objectives of this IPD-MA are a :
- To identify host, parasite and drug determinants of initial cure after the completion of treatment schedule.
- To identify host, parasite and drug determinants of definitive cure at 6 months after treatment completion.
- To identify host, parasite, and drug determinants of relapse at 6 months after treatment completion.
- To identify host, parasite, and drug determinants of mortality outcome at any time point during the study.

Primary and secondary endpoints relating to these objectives are defined in the outcomes section.

## METHODS AND ANALYSIS: PATIENTS, INTERVENTIONS AND OUTCOMES
### PICOT statement
*Population*: any patient enrolled in an interventional study with a confirmed or suspected diagnosis of VL defined by serological and/or parasitological testing.

*Interventions:* Any antileishmanial therapy

*Comparator:* Not restricted by the use of a comparator drug.

*Outcome:* at least one of the following outcomes reported: initial cure, definitive cure, relapse or mortality

*Time:* Studies published on or after 1980.

### Criteria for study eligibility
The following eligibility criteria must be fulfilled for study inclusion:
- Prospective clinical efficacy studies on patients with confirmed or suspected VL either using microscopy/serology/molecular method (ie, clinical diagnosis followed by a confirmatory method)
- Information on constituent drug(s), dose and duration of treatment regimen is available
- At least one of the following clinical outcomes are measured: initial cure, definitive cure, relapse or mortality

### Criteria for patient eligibility
The following minimum information are required for inclusion of a patient in the IPD meta-analysis:
- Details of antileishmanial treatment(s) administered
- Baseline information on age and gender
- Outcome is recorded

## OUTCOMES
### Outcomes and definitions
#### Primary endpoint
The primary endpoint adopted is cure at 6 months following treatment completion. This is commonly referred to as 'definitive' or 'final' cure. The precise definition and timing of definitive cure will vary according to study design. As a minimum, it is expected that a definitive cure is achieved if the patient remains alive and is no longer be manifesting symptoms and/or signs of VL.

Further analysis of this endpoint will be carried out using a stricter definition of definitive cure, defined as: achievement of initial cure (as defined below) and no subsequent rescue treatment or evidence of clinical or parasitological relapse, and no death associated with VL during the 6 months following treatment completion.

#### Secondary endpoints
The following secondary endpoints are adopted.

#### Initial cure
Initial cure is defined at the time of initial assessment adopted in the original study after completion of the therapy along with clinical improvement. The timing of initial cure assessment would typically take place within 28 days of treatment completion but this varies slightly across studies.

Further analysis of this endpoint will be carried out using a stricter definition of initial cure, defined as: cure at day 28 or at time-point defined in the original study following treatment, defined as clinical improvement of VL, absence of parasites in the spleen or bone marrow using microscopy and no rescue therapy on or before day 28 or before the timepoint of initial cure assessment adopted in the original study.

#### Relapse
Relapse is defined as the recurrence of signs and symptoms of VL at any time point during the study follow-up

among those who had achieved initial cure. The assessment typically takes place at 6 months but this varies slightly across studies.

Tissue aspirates for confirming parasitological presence are usually carried out based on clinical suspicion of relapse during the follow-up.[19 20] Since tissue aspiration for defining relapse would not be carried out in all the studies, further analysis of this endpoint will be undertaken by considering the methodology used to define relapse.

### Mortality
This is defined as deaths reported as related to the study interventions or due to the progressive worsening of the disease itself at any time during the study. Further analysis of this endpoint will be carried out by considering all-cause mortality.

### Variation is time-points
The assessment of initial cure and definitive cure is usually undertaken at 28-days and 180-days post-treatment. However, the exact time of these assessment will vary across the studies. Assessments undertaken between 15 and 60 days will be considered as time of initial cure. For assessments of definitive cure and relapse, assessments made between 150 210 days will be considered as 180 days.

## STATISTICAL METHODS
### Identification of relevant studies
Eligible studies were identified from the infectious diseases data observatory (IDDO) library of all prospective therapeutic studies that have systematically indexes all published therapeutic efficacy studies in VL published from 1980 onwards.[21] The IDDO VL clinical trials library is based on a living systematic review and the database is continually updated every 6 months in accordance with the Preferred Reporting Items for Systematic-Reviews and Meta-Analyses (PRISMA) guidelines. The trial library indexes publications identified from the following databases: PubMed, Embase, Scopus, Web of Science, Cochrane, clinicaltrials.gov, WHO ICTRP, Global Index Medicus, IMEMR, IMSEAR and LILACS. For this current review, the search includes all clinical trials published between 1 January 1980 and 2 May 2021. Details of the search strategy adopted are described elsewhere.[21] The search details are presented in a online supplemental file 1. Studies indexed in the IDDO VL library were eligible for inclusion in this review if they meet the inclusion and exclusion criteria outlined above. This review is not limited by language.

### Collation of IPD
Authors (principal investigators, corresponding authors) of the studies in the IDDO VL LSR library eligible for the inclusion in this IPD-MA were invited to be part of this collaborative research. Researchers agreeing to the terms

and conditions of the submission were then requested to upload IPD to the IDDO VL data platform through a secured web portal.[22] Data in the IDDO VL platform are fully pseudonymised to protect personal information and patient privacy.

### Data management
Raw data from individual studies shared with IDDO VL data platform are currently being standardised using the Clinical Data Interchange Standards Consortium (CDISC) compliant curation standards.[23] If required, investigators will be further contacted to clarify questions that arise during data curation and analysis, and individual study protocols will be requested. On standardisation, the data are stored in a relational database containing information on drug regimen, parasitological, clinical and haematological assessments, and therapeutic outcomes.

### Statistical methods for primary and secondary outcomes
#### Variables considered for regression modelling
The following variables will be considered for inclusion in the analysis of primary and secondary endpoints:

### Host variables
The following host variables are considered: age, gender, body weight, nutritional status, comorbidity status (such as HIV) and duration of illness prior to study enrolment. The following baseline clinical measurements will be considered: haemoglobin/anaemia status, spleen size, immunological biomarkers (CD4+ count), alanine transaminase (ALT) and aspartate transaminase (AST), platelets and neutrophils. If data are available, the following baseline characteristics will be considered: history of blood transfusion, bilirubin, creatinine, urea, and albumin concentrations.

### Parasite variables
The following parasite-related baseline factors will be considered: parasite load and information regarding the nature of infection (primary vs previously treated cases). Any cases described as previously untreated (or 'fresh') cases for leishmaniasis will be considered as primary VL. The enumeration of parasite density is usually carried out by evaluating bone marrow or splenic aspirates or through quantitation of parasites in unit quantity of specimen or parasite DNA equivalence per PCR reaction across studies;[24–27] the Chulay-Bryceson scale being the most commonly adopted.[24] The methodology used for parasite gradation will be considered in the analysis.

### Drug variables
The following drug-related variables will be considered for inclusion in analysis: mg/kg total dose (or target dose) administered,[28] treatment duration, whether the regimen was monotherapy or a combination therapy, mode of drug administration (intravenous, intramuscular or oral), administration of regimen either as a single dose or as a multiple doses, direct or indirect observations of drug administration for oral medications. The dosage

of antileishmanial interventions will be determined and expressed as dose per unit body weight (eg, total mg/kg dosage administered) and will be considered separately for each drug regimens.

### Study-level variables

The following study or arm level variables will be considered in the analysis of primary and secondary endpoints: country, study site, and calendar year of the study conduct.

### Geographical variation in treatment response

There is a known regional variation in treatment response in VL,[5] along with differences in patient characteristics and treatment guidelines. Therefore, a separate analysis will be undertaken within each geographical region to construct the univariable and multivariable regression models for the primary and secondary outcomes.

## STATISTICAL ANALYSIS AND REPORTING
### Descriptive summary

A summary of included studies will be presented with respect to study design, study location, year of study conduct, characteristics of study population, duration of follow-up, details of drug regimens including supervision of drug administration, and methodological details adopted for disease confirmation and treatment outcomes.

Summary of baseline characteristics of the patients included in the analysis will be presented by geographical region and overall. These include the following: age, weight, parasite grade at enrolment, temperature, haemoglobin (or haematocrit) concentration, anaemia and severe anaemia status, spleen size, description of severity of infection (severe/mild/uncomplicated) if available, total mg/kg dose for each treatment and supervision of drug administration. The distribution of the baseline characteristics will be summarised either as a proportion for categorical variables, as mean (with SD) or median (with IQR) for continuous variables.

### Analysis of the primary endpoint

A mixed effects logistic regression model will be used for identifying the risk factors for definitive cure in a one-stage IPD-MA. Given a relatively low number of expected events (for relapse) in each included study,[17] to avoid imprecise estimates from small events and continuity corrections, a one-stage IPD approach is preferred.[29] A random intercept regression model will be considered with study sites specified as random effects to account for within-site clustering of the patients (as some of the trials are multicentre studies).

If information regarding the time of occurrence of event is available, Kaplan-Meier (K-M) method will be used to estimate the probability of definitive cure at the end of the study follow-up, and a Cox regression model will be considered with study sites fitted as a frailty term to account for between-study site heterogeneity. The following outcomes will be censored in the survival analysis: lost to follow-up, death due

to causes adjudicated as unrelated to the study drug, voluntary withdrawal from the study.

### Multivariable modelling
*Core predictor set*

The following set of variables are the risk factors reported in clinical literature for treatment outcomes in VL: age, sex, baseline parasite density and HIV coinfection. These variables along with the drug regimen will form the minimal adjustment set for assessment of other risk factors and will be kept in the regression model regardless of statistical significance.[30]

### Assessment of other predictors and considerations for multivariable modelling

The association between the remaining candidate predictors and therapeutic outcomes will be assessed by adjusting for the core predictor sets identified. Multivariable model construction will follow the recommendations of Heinze et al.[31] Nested models will be compared by assessing the change in log-likelihood estimates and Akaike's information criterion will be used for comparing competing non-nested models. The functional form of the continuous variables will be determined using multivariable fractional polynomials[32] or restricted cubic splines.[30] Stability investigations will be undertaken to account for uncertainty introduced in multivariable modelling through bootstrap resampling.[31]

### Missing data

The proportion of missing observation for each of the variables considered in the analysis will be summarised and any missingness patterns will be explored. Multiple imputation,[33] which assumes missing at random mechanism for missingness, will be undertaken to handle missing observations. Construction of the imputation model will include all the variables in the target analysis (ie, all included exposures and outcome in the target analysis), and additional auxiliary variables including any interaction terms and non-linear associations. The number of imputations will be determined based on the fraction of missing information. The target analysis will then be carried out in each of the completed (observed plus imputed values) data sets and the estimates will be combined across the imputed data sets using Rubin's combination rules.[33]

### Subgroup analysis

Patients living with HIV who are treated for VL typically have worse outcomes and higher mortality risk than those who are not living with HIV.[34 35] Although, usually an exclusion criterion of therapeutic trials, a separate subgroup analysis will be carried out among patients with defined VL–HIV coinfections (data permitting). An interaction between treatment and HIV status (treatment–covariate interaction) will therefore be considered; within-study interaction will be separated from the between-study interaction by centring the covariates.[36]

## Sensitivity analyses

Stability investigations will be undertaken to account for uncertainty introduced in multivariable modelling through bootstrap resampling. The robustness of the derived estimates and their variance will be summarised using the recommendations in Heinze *et al.*[31] The influence of each of the studies towards the estimated regression coefficients will be assessed by removing each study at a time and estimating the coefficient of variation for the parameter estimates.

## Analysis of secondary endpoints
### Initial cure

The proportion of patients achieving initial cure (as defined previously) in each of the studies will be presented. The construction of univariable and multivariable regression models will follow the same approach as for the primary endpoint.

### Relapse

The proportion of patients achieving relapse (as defined previously) in each of the studies will be presented. The construction of univariable and multivariable regression models will follow the same approach as for the primary endpoint.

### Mortality

The proportion of deaths reported in each of the studies will be presented. The construction of univariable and multivariable regression models will follow the same approach as for the primary endpoint.

### Software

All analyses will be carried out using R software or Stata V.17 software.[37 38] Use of any other data analysis tools will not change the statistical analysis plan.

### Risk of bias assessment in included studies and in the IPD-MA

Risk of bias was assesed using the Cochrane risk of bias tool (RoB 2) for randomised studies.[39] Risk of bias in non-randomised studies will be carried out by assessing the preintervention, at intervention and postintervention domains as outlined in using ROBINS-I tool.[40]

To examine the risk of bias in IPD-MA, the first four domains of the quality in prognosis studies (QUIPS) tool and the first three domains of the prediction model risk of bias assessment tool (PROBAST) will be considered as recommended in Riley *et al.*[30] The relevant domains from the QUIPS checklist are study participation, study attrition, prognostic factor measurement and outcome measurement, and the relevant domains from PROBAST checklist are patient selection, prognostic factors and outcomes. Two reviewers will independently assess the risk of bias in the studies included in the analysis.

### Assessment of risk of potential bias in missing studies

Despite best possible efforts, it is anticipated that raw data from all the identified studies will not be available. The characteristics of patient population and study meta-data from the missing studies will be summarised to explore if the missing studies are systematically different from the studies that are included in the meta-analysis. A two-stage IPD may be conducted if sufficient details (or any covariate adjustment) are reported in the original studies.

## Dissemination plans
### Ethics and dissemination

This IPD-MA meets the criteria for waiver of ethical review as defined by the Oxford Tropical Research Ethics Committee (OxTREC) granted to IDDO, as the research consists of secondary analysis of existing anonymised data (Exempt granted on 29 March 2023, OxTREC REF: IDDO). Ethical approval was granted to each study included in this pooled analysis by their respective ethics committees. This IPD-MA will address research questions similar to that of included studies. Findings of this IPD-MA will be reported in open-access, peer-reviewed journals following the PRISMA-IPD guidelines.[41]

### Patient and public involvement

The research questions considered in this IPD-MA is based on a research agenda developed by the global VL research community.[42] The design and development of this IPD-MA were done by the study authors only and no patient was involved at any stage.

### Further development of statistical analysis plan

Major statistical analyses have been included in this plan. Amendments to the current plan or additional statistical analyses may be required as data accrual is in progress and will be transparently reported in subsequent reports of the results.

## DISCUSSION

Global commitment over the past few decades has led to significant progress in the control and elimination of VL. In the Indian subcontinent (ISC), the VL burden has greatly reduced in the past 15 years.[43 44] Availability of effective antileishmanial treatment is the cornerstone for achieving and sustaining elimination in the ISC. As the overall incidence of VL is falling, a growing proportion of the total reported cases in the region is attributed to relapses.[44] A recent report from South Sudan has also indicated increasing incidence of relapses over the past two decades.[45] This is an important public health concern as patients with relapse are predisposed to further relapse, especially among those with HIV coinfections.[12] Increasingly large proportion of VL patients has been found to present with HIV coinfections in Brazil (0.7% in 2001 to 8.5% in 2012), India (0.88% in 2000 to 4.19% in 2020) and Northern Ethiopia (15%–35%).[35] A study in India found that over half of those with HIV coinfections were unaware of their status.[46] This presents an important challenge to the ongoing control and elimination efforts as patients with VL-HIV coinfections typically have worse outcomes and higher mortality risk than those who are not

living with HIV.[34 35] VL-HIV patients are also recognised as an important reservoir of transmission as the coinfected patients are predisposed to multiple relapses, and have a high potential for infectiousness to sand-flies due to the generally high parasite loads and poses threat to the control and elimination efforts.[47 48] Therefore, the identification of host, parasite or drug-related characteristics remain crucial both for effective case management and also for disease control and elimination.

Several clinical trials and meta-analyses have highlighted the therapeutic efficacy and safety of one or more antileishmanial therapies.[5 11–13 16 34] However, risk factors determining the undesirable therapeutic outcomes remain poorly understood. Underlying heterogeneities in the published studies in terms of study population, treatment regimen, outcome definitions and study designs render difficulties in drawing a comprehensive conclusion regarding drug efficacy. In addition, the national treatment guidelines differ in terms of practices and approaches for case management. Some of these limitations could be addressed through a well-designed prospective study, which will incur a substantial logistical and financial costs to reach a critical mass required for robust investigation of the predictors of treatment relapse and remains unfeasible to achieve within short period of time. Utilising the existing data sets and undertaking a carefully planned IPD-MA can ameliorate some of these limitations.[18] We hope that this IPD-MA will provide evidence for the efficacy of different treatments in VL that can be further considered by policy makers at regional and global levels.

The identification of studies eligible for inclusion in the IPD-MA has been made through a comprehensive literature search of all published studies since 1980 with predefined inclusion–exclusion criteria. A major strength of this study is that data from several studies will be harmonised to a common standard based on an extensive consultation with the VL research community,[23] thus allowing us to address some of the methodological sources of heterogeneity.

A major challenge is that a substantial proportion of the studies in the IDDO library were conducted prior to the year 2000; the retrieval of data from historical trials is a major challenge. A particular scientific challenge is that misclassification of reinfection as 'true' relapse can potentially introduce bias in relapse estimates. A distinction of relapse and reinfection is seldom carried out in VL therapeutic trials and hence would not be considered in this IPD-MA. Evidence form a study in Nepal suggested late relapses were less likely to be new infections,[19] but generalisability to other settings remains unclear. Finally, the determinants of therapeutic outcomes in this IPD-MA would be limited to the commonly assessed and available clinical-laboratory parameters across VL drug trials. Similarly, the exploration of parasitic factors is limited to parasite gradation and the nature of the infection (primary vs previously treated cases). Other important parasite factors such as in vitro status of drug susceptibility, their virulence and the underlying genomic plasticity allowing parasites to undergo mutation under drug pressure[49 50] are not routinely collected in clinical trials and hence remains beyond the scope of this IPD-MA.

This IPD-MA will combine data from the studies in the IDDO VL data platform to identify the determinants of therapeutic outcomes. Findings of this research will generate important information for the control programmes at regional and global levels, policy-makers and groups developing new VL treatments.

**Author affiliations**
[1]Indian Council of Medical Research (ICMR)–Rajendra Memorial Research Institute of Medical Sciences (RMRIMS), Patna, India
[2]Infectious Diseases Data Observatory (IDDO), University of Oxford, Oxford, UK
[3]Centre for Tropical Medicine and Global Health, Nuffield Department of Medicine, University of Oxford, Oxford, UK
[4]Epidemiology and Communicable Diseases, Indian Council of Medical Research (ICMR), New Delhi, Delhi, India
[5]Infectious Disease Research Laboratory, Department of Medicine, Institute of Medical Sciences, Banaras Hindu University, Varanasi, India
[6]Drugs for Neglected Disease Initiative, Geneva, Switzerland
[7]Médecins Sans Frontières, Amsterdam, The Netherlands

**Twitter** Prabin Dahal @123prabindahal, IDDOnews, Abdalla Munir @AbdallaMunir_En and James Wilson @jamesemaj

**Contributors** Study conception: RK, PD, SS-P, NAS, AM, CN, JW, GB, MR, PM, FA, SS, KR, KS, PJG and KP. Project supervision: SS-P, MR, KP and PJG. Methodology: PD, NAS, SS-P, SS, KR, JW, FA, PJG and KS. Data curation: AM, SS-P, PM, JW, PD and GB. Project administration: SS-P and CN. Funding acquisition: PJG. Resources: SS-P, FA and PJG. Writing-original draft: RK, PD, KP, PJG and KS. Writing-review and editing: all authors were involved in reading and critical revision of the initial draft and approved the final manuscript.

This work is funded by a Bill & Melinda Gates Foundation grant to the Infectious Diseases Data Observatory, Oxford University, UK (Recipient: Prof. Philippe Guerin; ref: INV-004713). Funding agency had no role in developing the protocol.

**Competing interests** None declared.

**Patient and public involvement** Patients and/or the public were not involved in the design, or conduct, or reporting or dissemination plans of this research.

**Patient consent for publication** Not required.

**Ethics approval** Not applicable.

**Provenance and peer review** Not commissioned; externally peer reviewed.

**ORCID iDs**
Rishikesh Kumar http://orcid.org/0000-0001-8993-4683
Prabin Dahal http://orcid.org/0000-0002-2158-846X
Sauman Singh-Phulgenda http://orcid.org/0000-0003-2892-3053
Abdalla Munir http://orcid.org/0000-0003-2994-0154
James Wilson http://orcid.org/0000-0003-3615-4928
Krishna Pandey http://orcid.org/0000-0001-5930-2458

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
