## [Reviewer comments · BMJ Open]

ARTICLE DETAILS

TITLE (PROVISIONAL)	Host, parasite, and drug determinants of clinical outcomes following treatment of visceral leishmaniasis: a protocol for individual participant data meta-analysis
AUTHORS	Kumar, Rishikesh; Dahal, Prabin; Singh-Phulgenda, Sauman; Siddiqui, Niyamat; Munir, Abdalla; Naylor, Caitlin; Wilson, James; Buck, Gemma; Rahi, Manju; Malaviya, Paritosh; Alves, Fabiana; Sundar, Shyam; Ritmeijer, Koert; Stepniewska, Kasia; Guérin, Philippe J.; Pandey, Krishna

VERSION 1 – REVIEW

REVIEWER	Doni, Shimelis Alberta Health Services
REVIEW RETURNED	09-Jun-2023

GENERAL COMMENTS	1. One of the critical element in the treatment outcome measurement is the experience of treating VL using different treatment guidelines across India subcontinent and east Africa and this should be addressed in detail in the discussion part of the manuscript.2. Ethical approval might be required since you are planning to request amendment if necessary.3. Hiv coinfection is one of the major predictor of treatment outcome and that needs further discussion in the manuscript.4. there are many limitations in your study that could affect the results of your study and some of them could be answered by conducting cohort study.
--

REVIEWER	Jeffares, Daniel University of York
REVIEW RETURNED	20-Jun-2023

GENERAL COMMENTS	This is a timely and valuable study. Within the limitations of the data that is available, it is likely that the current state of VL clinical trials will be summarised very well by the meta analysis proposed, with potential for novel insight. A limitation I observe is that I cannot see how parasite determinants of outcome can be evaluated with the data that is available at IDDO, since most studies do not explicitly contain any data on parasites. While it would be possible to make inferences that differences between parasites within and between continents have effects, these are very much confounded with other factors that vary with the same geographic range - such as environmental, socio-economic and host genetic or infection differences.
--

	Given this I would suggest a critical evaluation of the extent to which you can detect parasite determinants.
REVIEWER	Gorgani-Firouzjaee, Tahmineh Babol University of Medical Science
REVIEW RETURNED	15-Jul-2023
GENERAL COMMENTS	This review provide comprehensive data on VL treatment protocols. Please add ethical approval cod/number in the text.

VERSION 1 – AUTHOR RESPONSE

Reviewer: 1 Dr. Shimelis Doni, Alberta Health Services

Comment #1. One of the critical elements in the treatment outcome measurement is the experience of treating VL using different treatment guidelines across India subcontinent and east Africa and this should be addressed in detail in the discussion part of the manuscript.

Authors' response: We would like to thank Dr. Doni for this important comment as we have mainly focused on the patient, parasite and drug factors that can influence the therapeutic outcomes, and overlooked this important aspect of disease epidemiology. The regional variation in the drug efficacy, underlying state of parasitic resistance and patient characteristics has meant that different first line therapies has been adopted across the regions. We have acknowledged this and plan to address this source of variability through separate analysis within each geographical region.

We have made the following revisions:

Lines 265-269:

There is a known regional variation in treatment response in VL [5], along with differences in patient characteristics and treatment guidelines. Therefore, a separate analysis will be undertaken within each geographical region to construct the univariable and multivariable regression models for the primary and secondary outcomes.

Lines 417-421:

Underlying heterogeneities in the published studies in terms of study population, treatment regimen, outcome definitions, and study designs render difficulties in drawing a comprehensive conclusion regarding drug efficacy. In addition, the national treatment guidelines differ in terms of practices and approaches for case management.

Comment #2. Ethical approval might be required since you are planning to request amendment if necessary.

Authors' response: We have obtained ethical approval for this secondary analysis of pseudonymised data. Any deviations from the target analysis outlined in this protocol will be clearly and transparently communicated in the subsequent reports/manuscripts detailing the results arising from this analysis.

The following changes are made on Lines 390-393:

Amendments to the current plan or additional statistical analyses may be required as data accrual is in progress and will be transparently reported in subsequent reports of the results.

Comment #3. HIV coinfection is one of the major predictors of treatment outcome and that needs further discussion in the manuscript.

Authors' response: We have also further highlighted in the manuscript on lines 403-414:

Increasingly large proportion of VL patients have been found to present with HIV co-infections in Brazil (0.7% in 2001 to 8.5% in 2012), India (0.88% in 2000 to 4.19% in 2020) and Northern Ethiopia (15-35%)[35]. A study in India found that over half of those with HIV co-infections were unaware of their status [46]. This presents an important challenge to the ongoing control and elimination efforts as patients with VL-HIV co-infections typically have worse outcomes and higher mortality risk than those who are not living with HIV [34,35]. VL-HIV patients are also recognised as an important reservoir of transmission as the co-infected patients are predisposed to multiple relapses, and have a high potential for infectiousness to sand-flies due to the generally high parasite loads and poses threat to the control and elimination efforts [47,48]. Therefore, the identification of host, parasite or drug related characteristics remain crucial not only for effective case management but also for disease control and elimination.

Comment #4. There are many limitations in your study that could affect the results of your study and some of them could be answered by conducting cohort study.

Authors' response: We agree that a prospective study (randomised or cohort) will alleviate several of the limitations identified (a retrospective cohort study will still have some of the limitations). Among treated patients, approximately 5% of patients eventually relapse (among non-HIV patients). Therefore, to undertake a robust analysis to identify the predictors of relapse, such prospective cohort study will require a large sample size. This will incur a substantial logistical and financial costs and remains unfeasible to achieve within short period of time. The main strength of our IPD-MA is in numbers as collation of dataset from several trials will increase the effective sample size (the number of events) thus allowing a robust exploration of the predictors and maximising the utility and re-use of existing datasets.

The following statement is added on lines 421-426:

Some of these limitations could be addressed through a well-designed prospective study, which will incur a substantial logistical and financial costs to reach a critical mass required for robust investigation of the predictors of treatment relapse and remains unfeasible to achieve within short period of time. Utilising the existing datasets and undertaking a carefully planned IPD-MA can ameliorate some of these limitations [18].

Reviewer2: Dr. Daniel Jeffares, University of York

Comments to the Author: This is a timely and valuable study. Within the limitations of the data that is available, it is likely that the current state of VL clinical trials will be summarised very well by the meta-analysis proposed, with potential for novel insight.

A limitation I observe is that I cannot see how parasite determinants of outcome can be evaluated with the data that is available at IDDO, since most studies do not explicitly contain any data on

parasites. While it would be possible to make inferences that differences between parasites within and between continents have effects, these are very much confounded with other factors that vary with the same geographic range - such as environmental, socio-economic and host genetic or confection differences. Given this I would suggest a critical evaluation of the extent to which you can detect parasite determinants.

Authors' response: We would like to thank Dr. Jeffares for acknowledging the importance of our planned study and also for raising this important point regarding parasite related characteristics. We agree that the collection of data on parasite-related characteristics remains relatively limited to enumeration of parasite load and the nature of infection (primary vs previously treated cases). While the latter is reported at study level made explicit as a part of the study eligibility criteria, latter (parasite density estimation) is reported in most trials since confirmation of the disease requires demonstration of parasites in the tissue aspirate. However, we appreciate that not all studies that undertake tissue aspirate might explicitly enumerate the parasite density. As stated in the methods section, we will account for this geographical variation by undertaking separate analysis within each region.

The following limitations is now acknowledged on lines 443-448:

Similarly, the exploration of parasitic factors is limited to parasite gradation and the nature of the infection (primary vs previously treated cases). Other important parasite factors such as in vitro status of drug susceptibility, their virulence, and the underlying genomic plasticity allowing parasites to undergo mutation under drug pressure[49,50] are not routinely collected in clinical trials and hence remains beyond the scope of this IPD-MA.

Reviewer3: Dr. Tahmineh Gorgani-Firouzjaee, Babol University of Medical Science

Comments to the Author: This review provides comprehensive data on VL treatment protocols. Please add ethical approval code/number in the text.

Authors' response: This IPD-MA meets the criteria for waiver of ethical review as defined by the Oxford Tropical Research Ethics Committee (OxTREC) granted to IDDO, as the research consists of secondary analysis of existing anonymised data. The exemption was granted on 29th march 2023 and the approval code has IDDO as the application reference. This has been added in the revised manuscript in lines 378-381:

"This IPD-MA meets the criteria for waiver of ethical review as defined by the Oxford Tropical Research Ethics Committee (OxTREC) granted to IDDO, as the research consists of secondary analysis of existing anonymised data (Exempt granted on 29th March 2023, OxTREC REF: Infectious Diseases Data Observatory (IDDO))."